# A Meta-Analysis of the Effectiveness of High, Medium, and Low Voltage Electrical Stimulation on the Meat Quality of Small Ruminants

**DOI:** 10.3390/foods9111587

**Published:** 2020-11-02

**Authors:** Archana Abhijith, Frank R. Dunshea, Robyn D. Warner, Brian J. Leury, Minh Ha, Surinder S. Chauhan

**Affiliations:** 1School of Agriculture and Food, The University of Melbourne, Parkville, VIC 3010, Australia; archana.abhijith@student.unimelb.edu.au (A.A.); fdunshea@unimelb.edu.au (F.R.D.); robyn.warner@unimelb.edu.au (R.D.W.); brianjl@unimelb.edu.au (B.J.L.); minh.ha@unimelb.edu.au (M.H.); 2Faculty of Biological Sciences, The University of Leeds, Leeds LS2 9JT, UK

**Keywords:** electrical stimulation, meat color, small ruminants, tenderness, water holding capacity

## Abstract

The current study is a meta-analysis of the effects of electrical stimulation (ES, *n* = 28 papers) with different voltages combined with different ageing periods (0–3, 4–7, and 8–14 days) on the meat quality of small ruminants. A comprehensive search for published studies on meat quality of small ruminants investigating the application of low, medium, and high voltage electrical stimulation, was performed using Google Scholar, ScienceDirect, PubMed, and Scopus databases. Forest plots, funnel plots, and other statistical tools and tests were used in the study to analyze the results. Electrical stimulation significantly reduced ultimate pH (*p* < 0.001), Warner–Bratzler shear force (WBSF) (*p* < 0.001), cooking loss (*p* < 0.05), and purge loss (*p* < 0.001). In addition, sarcomere length (*p* < 0.01), myofibrillar-fragmentation index (MFI) (*p* < 0.001), and color (L*, a*, b*) (*p* < 0.001) showed higher values in meat subjected to ES as compared with the control group. In conclusion, the meta-analysis revealed statistical proof of beneficial effects of ES on meat quality of small ruminants in terms of ultimate pH, tenderness, enhanced proteolysis, and higher colorimetric values.

## 1. Introduction

The increasing consumer demand for high quality and minimally processed meat has resulted in increased focus on food processing technologies for the meat industry. Among these technologies, electrical stimulation (ES) is a technique practiced in many commercial meat processing plants globally to improve tenderization in beef, sheep, and lamb [1,2]. Electrical stimulation technology involves passing an electric current through the carcass of freshly slaughtered animals [3]. This causes the muscle to contract, resulting in an increase in glycolysis and a rapid decline in pH. This technology was first used commercially in New Zealand to accelerate the onset of rigor mortis [3]. Electrical stimulation has become an important component of sheep meat processing, with the benefits of reducing inconsistency in eating quality [4,5], as well as increasing the number of carcasses reaching the ideal pH temperature window [6]. Cold shortening/toughening that happens due to rapid chilling of pre-rigor muscles has led to the development of this industrial technology [7]. Electrically stimulated muscles enter rigor before the muscle temperature is conducive to pre-rigor shortening [3]. Although the decline in pH and improvement in meat tenderness as a result of electrical stimulation was unequivocal in most published studies, equivocal responses were observed on other traits such as color stability [8,9]. Compared to beef and sheep meat, little is known about the effect of electrical stimulation on goat meat.

As well as electrical stimulation, post-mortem ageing is well established as an important strategy to improve tenderness of meat [10]. In general, meat tenderization is mainly caused by ultrastructural changes that weaken the integrity of the myofibers in the muscle tissue [11]. In addition, meat ageing also causes changes in water holding capacity and color. While few studies have examined the effect of electrical stimulation and ageing individually and interactively on meat quality, a comprehensive analysis, such as a meta-analysis, is needed to obtain a more accurate assessment of the effectiveness of these processing technologies for the meat industry.

Meta-analysis is a powerful statistical tool extensively utilized in biomedical and clinical research [12,13]. Furthermore, exploration of the sources of heterogeneity in a meta-analysis can provide additional information on some of the factors that contribute to the variation in response to an intervention or treatment [14]. While meta-analysis is a common tool in biomedical science, its use in meat science research has been limited. A few recent studies in the field of science have used meta-analysis to examine the effects of various treatments, for example, the works on understanding the effects of various metabolic modifiers and different processing technologies on meat and carcass quality [15,16,17,18,19]. This meta-analysis aims to evaluate the effect of electrical stimulation on small ruminant meat quality, using published data. Heterogeneity or variability of responses in individual studies on same traits were also examined in order to identify contributors to variations in meat quality and resolve the current gaps in knowledge.

## 2. Materials and Methods

### 2.1. Literature Search and Data Collection

A comprehensive search for published studies on meat quality of small ruminants investigating the application of low, medium, and high voltage electrical stimulation, was performed using Google Scholar, ScienceDirect, PubMed, and Scopus database. Key words used for searches were electrical stimulation, low, medium, or high voltage stimulation or all three in combination with meat and with species.

### 2.2. Paper Inclusion/Exclusion Criteria

The following inclusion criteria were used to build a database in research papers: full text manuscripts from peer-reviewed journals; species and muscles were clearly stated; specific voltage applied with duration of treatment and time of treatment post-mortem were stated; ageing time was provided if applied, papers providing adequate data for determining the effect size of treatment outcomes; and procedures of the selected parameters were clearly described. Papers were excluded from the study if they did not meet all the required criteria. The mean values of control and treatment groups, the number of samples allocated to control and treatment, and a measure of variance expressed either as standard error (SE) or standard deviation (SD) were kept as the required data that were noted from all the selected papers for the meta-analysis. The papers comprising the database are listed at the bottom of Table 1.

### 2.3. Database

The database comprises author names with year; species; where possible breed, age, weight, and ultimate pH; time of application with current, frequency, and pulse interval, as well as ageing time if applied; and treatment parameters for electrical stimulation (low, medium, or high voltage). The data were exclusively on the meat quality of small ruminants including lamb, sheep, and goat on the longissimus muscle. For a better understanding of the effects to specific groups, in addition to different species, the study also considered commercial categories of the same species such as lamb and sheep. The methodology criteria used for the database was as follows: Warner–Bratzler shear force (WBSF) and cooking loss (CL) samples were cooked in a water bath for 30 min until the core temperature reached 70 °C. The CL, defined as the amount of weight loss of the samples during cooking, was calculated as (raw weight – cooked weight)/raw weight × 100 and presented as a % change. Color measurement in all the papers was performed using a Minolta or HunterLab Miniscan instrument and expressed as L* (lightness), a* (redness/greenness), and b* (yellowness/blueness) [20]. Purge loss was measured as the percentage weight loss in vacuum-packed meat samples and weighed at allocated ageing days. Muscle samples assigned for sarcomere length and myofibrillar fragmentation index (MFI) were frozen at −20 °C. Sarcomere length was determined using a helium-neon laser diffraction unit. The selection of these parameters used, depended on the published scientific reports on electrical stimulation effects on meat quality. Meat pH, WBSF, cooking loss, purge loss, sarcomere length, MFI, and meat color are the major objective meat quality characteristics that have been uniformly studied across all the individual studies.

### 2.4. Statistical Analysis

The meta-analysis was conducted, based on procedures previously reported [17], using Review Manager (RevMan) software version 5.3 (The Nordic Cochrane Centre, The Cochrane Collaboration, Copenhagen, Denmark, 2014). The effect of electrical stimulation on the meat quality parameters was computed using the standardized mean difference (also known as effect size), following the method described by Vesterinen et al. [21]. A measure of variance expressed as SD (SD calculated from SE and sample size) for each group was obtained for all the parameters. The mean values of control and treatment groups along with the reported values of SD or SE were populated in the software database, which automatically compiled the effect size and other statistical test outcome. The effect size was computed using both fixed and random effects models with the overall effect size estimates, 95% confidence intervals, and overall test possibilities (P) reported. The entire data were split into subgroups and each subgroup was analyzed individually. The six subgroups assigned in the study included 3 voltage intensities, i.e., high voltage, medium voltage, and low voltage. In this study, voltage settings above 250 V were considered to be high voltage ES and above 100 V were considered to be medium voltage ES. For the purpose of this study, ES that fell below the medium voltage setting was considered to be low voltage ES. Ageing was identified as the major factor that had been undertaken in most of the identified papers. However, the duration was not uniform and was not undertaken in certain studies, thus, these were treated as the following different subgroups: 1–3 days, 4–7 days, and 8–14 days of post-mortem ageing. The age of lambs ranged between 6 and 8 months, sheep 18–24 months, and goat kids 12–48 months. The mean hot carcass weight and SD (in reported studies) of the animals were 23.8 ± 6.2 kg (22.6–24.8) for sheep, 18.5 ± 4.2 kg (15.6–20.5) for lambs, and 12.3 ± 2.3 kg (10.1–14.2) for goat kids.

The effect size estimates and the corresponding 95% CI of the effect size of individual experiments and the overall effect were analyzed with the help of forest plots. Outliers were excluded from the study at this stage using the forest plot. Chi-square test and *I*^2^ (*I*^2^ = 100 × (Q − d.f.)/Q, in which Q is the Cochran’s heterogeneity statistic and d.f. the degrees of freedom) were used to assess heterogeneity in effect size of the treatment outcome [22]; *I*^2^ <30% was considered to be mild heterogeneity, 30–50% as moderate, and >50% as severe heterogeneity [22]. This means that only *I*^2^ was the true heterogeneity and the remaining was due to the sampling errors of individual studies. Asymmetry of the funnel plots was used to assess publication biases [23]. A random effects model was used for pooling effect estimates because effect sizes from animal studies were more likely to differ due to the difference in design characteristics [24]. The tests for funnel plot asymmetry were not used if the study number was less than 10, because the test power is generally too low to differentiate chance from real asymmetry [25].

## 3. Results and Discussion

The internet search resulted in 28 studies on ES in small ruminants, which met the selection criteria for the meta-analysis (Table 1). All the studies were published between 1976 and 2014. The meta-analysis included 16 lamb, nine sheep, and six goat studies. All responses are expressed as effect size. The rule of thumb used for effect size was small for effect size >0.2 and <0.5; medium for effect size >0.5 and <0.8; and large for effect size >0.8 [26]. High voltage ES involves application of a fixed current across the whole carcass at the completion of the dressing procedure, 20–30 min after dressing with 1130 V peak, the r.m.s. voltage (root-mean-squared, effective value of alternating current) is 800 V, and peak current 1.1 A with a 14 Hz sinusoid waveform applied for 100 s [27]. Although high voltage ES can significantly reduce toughness in sheep meat, the high voltage process poses higher risk for workers safety. Consequently, the medium voltage ES was introduced later [10] in compliance with the occupational health and safety regulations in Australia (Australian Standard 60479-2002) [28], which can be applied to a carcass during both pre- and post-dressing. In this system, the current remains constant, and the voltage is varied with a peak of around 300 V, by an electronic controller that determines the electrical resistance from the carcass. This feedback system adjusts the voltage accordingly. It is usually performed within 2–5 min after stunning. Pearce et al. [29] compared different combinations of current and pulse widths in a dressed lamb carcass. They showed that the combination of 2.5 ms and 1000 mA was the most effective treatment with respect to both Sheep Meat Eating Quality (SMEQ) and Meat Standards Australia (MSA) guidelines. The highest number (60%) of carcasses reached the SMEQ window which was temperature, at a pH of 6, between 18–25 °C and the MSA window (97%) which was temperature, at a pH of 6, between 15–35 °C. Later in 2009, a study was conducted on alternating frequency to increase response from medium voltage ES in lamb carcasses [30]. The authors showed that maintaining a constant frequency of 15 Hz with pulse width of 2.5 ms and 1000 mA current resulted in a higher number of carcasses achieving the required temperature, at a pH of 6, between 18–25 °C and improved tenderness.

### 3.1. Effect on Ultimate pH

When all the available data were combined, the meta-analysis was able to detect a medium effect size (−0.61) but statistically significant decrease in ultimate pH of −0.03 (Table 1). The negative value of effect size indicates the lower ultimate pH in ES group as compared with the control (NES, non-electrically stimulated). Comparing the individual effect size of high (−0.72), medium (−0.74), and low (−0.39) voltage ES, high and medium showed high effect size and low voltage showed a smaller effect size. However, the heterogeneity value (*I*^2^) (variation in the outcome values between studies) was 0% for the low voltage group, which was considered to be the most reliable result. Other subgroups hold moderate heterogeneity. The statistical significance of the combined (*p* < 0.001) and individual effect of high (*p* < 0.001), medium (*p* < 0.001), and low (*p* < 0.001) voltage effect of subgroups of electrical stimulation might be interpreted as evidence or statistical proof for the effect of electrical stimulation on ultimate muscle pH.

The underlying mechanisms for the effects of electrical stimulation lies in its ability to hasten the onset of rigor mortis through rapid post-mortem muscle glycolysis, followed by an increased rate of pH decline [31]. While the effect of ES on ultimate pH is not unexpected, the study conducted by Toohey et al. [32] comparing high, medium, and low voltage provided much needed evidence to drive a change in the industry. The success of the medium voltage stimulation was in the flexibility of this system to be administered on a wool-on carcass and the worker safety and feasibility of installation in the slaughter chain.

### 3.2. Effect on Warner–Bratzler Shear Force (WBSF)

The effect of ES on WBSF is presented in Table 1. A medium effect size (−0.70) but highly significant (*p* < 0.001) decrease in WBSF of −1.04 kg was detected. However, there was severe heterogeneity (78%) indicating that there are several other factors impacting WBSF. When comparing the effect size of subgroups, ES meat aged for 1–3 days showed the greatest reduction in WBSF (−0.78) as compared with the other subgroups. Most of the studies in the database have reported a reduction in WBSF in meat subjected to ES followed by meat ageing [1,32,33,34,50], although in approximately half the comparisons, the effects were statistically insignificant (Figure 1). Thus, using the statistical power of a meta-analysis, we demonstrated that ES was effective for improving tenderness of small ruminant meat.

A forest plot was constructed to understand the effect of ES on WBSF in different species (Figure 1). The diamond in the figure indicates the overall effect estimate (−0.70) which clearly shows reduction in WBSF in ES meat, overall as well as in sheep (−0.86), lamb (−0.43), and goats (−0.59) (*p* < 0.01). Although the difference between ES and NES meat was reported as significant in most of the studies, it can be seen in the forest plot that only half of the studies showed a significant effect when a meta-analysis was performed. Figure 2 illustrates that there is publication bias in the meta-analysis, with the asymmetry of the funnel plot indicating that studies with no beneficial effects are missing [51]. Although funnel plot asymmetry has long been equated to publication bias, it also underpins the small-difference studies, which is a tendency of small studies to have different intervention effects as compared with large study effects [25]. Smaller studies with lower sample size usually contribute less to a meta-analysis [23]. True heterogeneity is another reason leading to funnel plot asymmetry [25]. In this study, disparities in the experimental settings (high, medium, low voltage, varying frequency, and pulse widths); ageing period (shorter to longer duration); and age group could be some of the several factors contributing to the heterogeneity as all of these have a large role in meat tenderness. Due to the lack of studies, the meta-analysis could not differentiate these factors.

Tenderness remains a crucial determinant of eating quality. An underlying mechanism by which ES improves meat tenderization is prevention of cold shortening, when the temperature drops below 10–12 °C. The underlying mechanism of cold shortening appears to occur from the inability of sarcoplasmic reticulum and mitochondria to sequester calcium ions at temperatures below 10 °C, which causes an excess of calcium ions in the intracellular space which activates muscle contraction. Adequate ATP levels need to be present in muscle for cold shortening to occur. The result of cold shortening is shortened sarcomeres leading to meat toughness [31]. Dutson et al. [52] showed the release of cysteine proteases, such as cathepsins, in an electrically stimulated lamb carcass, which has the potential to degrade the myofibrillar proteins [53].

### 3.3. Effect on Sarcomere Length

Meat subjected to ES had a medium effect size (0.45) with an increase in sarcomere length of 0.01 μm as compared with the NES group (*p* < 0.01, Table 1). There was severe heterogeneity (74%), and when comparing the subgroups, low voltage ES had the highest increase in sarcomere length as compared with the NES meat.

Although the results showed that sarcomere length tended to increase in ES meat, Toohey, et al. [32] explained this effect of ES on sarcomere length to be a function of the chilling regimen. This assumption was supported by a previous study done by White et al. [54], where carcass held at 5 °C had significantly shorter sarcomeres than samples held at 15 °C. However, some studies reported shorter sarcomeres as a result of intense ES in lamb [35]. Generally, cold-induced shortening in muscle is associated with shorter sarcomeres and an overlap of thick (myosin) and thin (actin) myofilaments, which results in meat toughness [55,56]. Ertbjerg and Puolanne [56] reviewed the marked effects of sarcomere lengths on textural properties of raw and cooked meat, and conclusively stated the market influence of sarcomere length on water-holding capacity and indirect effects on color. There are disparate views on the relation of sarcomere length to meat tenderness. While some showed the direct correlation of sarcomere length to tenderness [57], others showed the effect of ageing for improving shortened muscles in lambs [58,59]. The ultimate shear force value of meat is associated with the sarcomere length and the extent of postmortem proteolysis [60] but, the interaction between these elements is still not clear. However, several studies have shown longer sarcomere length to be associated with the faster degradation rates of myofibrillar proteins such as troponin-T [55,61] and titin [62] in beef.

### 3.4. Effect on Cooking Loss (CL)

Meat subjected to ES showed a small (-0.11) but significant decrease (*p* < 0.05) in CL of –0.03% as compared with NES meat (Table 1). Highly reliable heterogeneity (0%) indicates statistical evidence of reduction of CL in ES meat. When comparing the different ageing groups, the effect size gradually increased with the ageing period, showing a positive relationship between the ageing period and reduced CL. Some of the studies, however, showed no effect of ES on cooking loss in sheep and goat [2,33,36]. In contrast to this study, water holding capacity was reported to be lower in ES meat as compared with NES in bulls. This was explained to be the result of the denaturation of hydrophilic proteins [63].

### 3.5. Effect on Purge Loss

Meta-analysis for purge loss was done using only four studies, which showed a highly significant (*p* < 0.001) but medium size (−0.61) decrease of purge loss in meat subjected to ES as compared with NES meat (Table 1). Heterogeneity was not considered, as the study number was too small.

Although the underlying mechanism relating meat proteolysis and water holding capacity is not well understood, the association of increased proteolytic activity and higher water holding capacity (reduced drip, purge, and cooking loss) has been well-established [64,65]. The authors suggested a model in which the myofibrillar protein degradation caused a change in the myofibrillar structure creating a swelling that allowed water to flow from the extracellular to the intracellular space.

### 3.6. Effect on Meat Color (L*, a*, b*)

ES had an effect on meat color (Table 1). The meta-analysis showed increase (*P* < 0.001) of 0.16, 0.11, and 0.08 in lightness (L*), redness (a*), and yellowness (b*) values, respectively, of meat subjected to ES as compared with NES meat (effect estimates, L* 0.59, a* 0.40, b* 0.34). Despite the small number of studies, these results are considered to be reliable as the heterogeneity of both a* and b* were 0%. The L* value had severe heterogeneity of 62%.

Lower L* and higher a* values usually results in darker colored meat, which is associated with high ultimate pH of meat [2,66]. This agrees with our meta-analysis results for the higher lightness (L*) and redness (a*) values which could be associated with the lower ultimate pH of ES meat as compared with NES meat. Lawrie [66] suggested that the fast fall in pH, due to ES, caused the muscle proteins to approach their isoelectric point much faster, which opened the structure allowing oxygenation of myoglobin. Previously, the association of high ultimate pH meat with dark cutting was explained by Hughes et al. [67] as a result of the swelling of the fibers, which resulted in reduced space between muscle fibrils, causing decreased light scattering.

### 3.7. Effect on Myofibrillar Fragmentation Index (MFI)

As presented in Table 1, there was a significant (*p* < 0.01) increase in MFI of 0.01 in ES meat (effect estimate 0.22). There was mild heterogeneity of 11% which indicated the reliability of these results. MFI is considered to be a potential indicator of the extent of proteolysis related to the rupture of the I-band and breakage of intermyofibrillar linkages [68]. Higher MFI in ES meat followed by ageing was linked with reduced shear force and improved tenderness and has been reported extensively in sheep and lamb [4,34,69]. The role of rapid pH decline during ES in enhancing the muscle proteolysis and fiber disruption and increasing tenderization in meat has been reported in some studies [1,33]. Martin et al. [34], in an attempt to understand the role of a structural protein (desmin) and a regulatory protein (Troponin-T) in meat tenderization following electrical stimulation in lambs, showed higher degradation rates of these proteins in electrically stimulated muscles. However, the same study reported a strong correlation between undegraded desmin and high shear force values at a pH of 5.8. The authors explained this as the high rigidity of myofibrer which made the meat tough. Contreras-Castillo et al. [70] showed that in bull beef, the degradation rate of myofibrillar proteins including titin, desmin, and troponin-T was faster with ES, coinciding with increased tenderization of meat. Although ES is not a new initiative, detailed studies on understanding the effect of electrical inputs on the meat tenderization pathway will provide in-depth insights on the fundamental mechanisms.

## 4. Conclusions

Meta-analysis is a useful technique to comprehensively assess the effect of electrical stimulation and ageing on meat quality from a number of studies, while minimizing the discrepancies from individual studies. The findings from the current study provide strong evidence for the positive effects of electrical stimulation in meat quality of small ruminants. High heterogeneity in some parameters may be due to differences in experimental designs, breed, sex, and other environmental factors, and experimental settings. On the basis of the overall effect estimates computed in this meta-analysis, it can be concluded that electrical stimulation increases ultimate pH, WBSF, sarcomere length, and MFI, and decreases cooking loss and purge loss in a carcass. Overall, this meta-analysis underpins the effectiveness of the combination of electrical stimulation with meat ageing for improving meat tenderness. One of the main purposes of electrical stimulation is to reduce the occurrence of cold shortening in a carcass. Hitting the ideal pH temperature window can make the meat achieve acceptable tenderness, reduce the variation in tenderness, and enhance meat color. Although there have been several scientific reports establishing the effects of electrical stimulation, this meta-analysis could be considered as a statistical proof for the adoption of this technology by processors. The results of the study need special attention, as the study has considered subgroup analysis (different species, voltages, and ageing periods) which could help us to understand the effects of this technique on specific groups rather than considering a broad population. This also reduces the overestimation/underestimation of the overall effect that usually occurs while considering a large population. Electrical stimulation is practiced commercially in several processing plants. Medium voltage electrical stimulation is a proven safe and feasible installation in the meat slaughter chain, especially considering the feasibility of this system for stimulating sheep with a wool-on carcass. Toughness of meat from small ruminants (sheep, goat, and lamb) is a critical determinant of repurchase decision by consumers, especially in western countries. The benefits of electrical stimulation on meat quality should be promoted in commercial plants to improve eating quality consistency in the supply chain.

## Figures and Tables

**Figure 1 foods-09-01587-f001:**
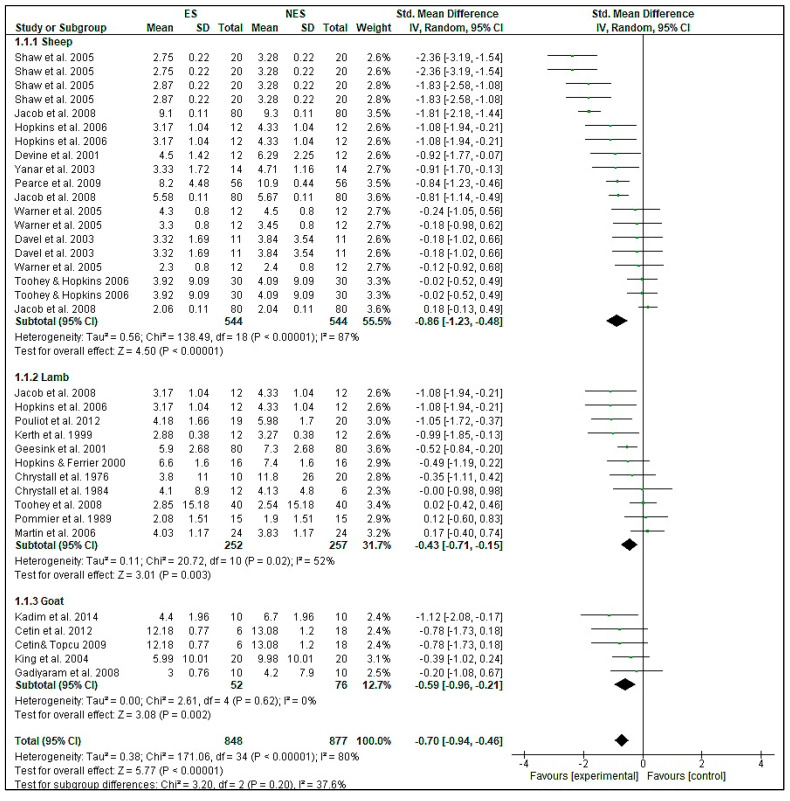
Forest plot of Warner–Bratzler shear force (kg) responses in electrical stimulation studies. A forest plot was used for the graphical display of the estimated results/effect size of the effect of electrical stimulation on Warner–Bratzler shear force, using the standardized mean difference (standardized using the z-statistic) and 95% confidence interval. The weights that each study contributed are presented. The solid vertical line represents a mean difference of zero or no effect. Points to the left of the line represent a reduction in Warner–Bratzler shear force, while points to the right of the line indicate an increase. The upper and lower limit of the lines connected to the square (effect size) denotes the 95% confidence interval for the effect size. The overall effects size for all species with 95% confidence interval is indicated by the diamond at the bottom. This effect was heterogenous, as indicated by the *I*^2^ of 80.0%.

**Figure 2 foods-09-01587-f002:**
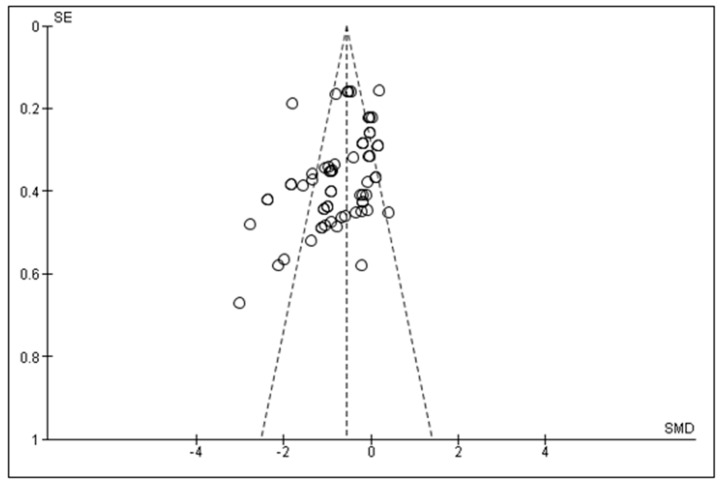
An asymmetrical funnel plot with the publication bias showing the effects of electrical stimulation on Warner–Bratzler shear force (kg) in sheep, lamb, and goats. The gap in the bottom indicates missing smaller studies with no beneficial effects. SMD and SE indicate standardized mean difference and standard error.

**Table 1 foods-09-01587-t001:** Meta-analysis results of effects of electrical stimulation (high, medium, and low voltage) and effect of different ageing duration (1–3 days, 4–7 days, and 8–14 days) combined with electrical stimulation on ultimate pH; Warner–Bratzler shear force (WBSF, kg); sarcomere length (μm); cooking loss (% CL); purge loss (%); L*, a*, b* values; and myofibrillar fragmentation index (MFI) of meat from small ruminants. CI = confidence interval, *I*^2^ = percentile of total variation due to heterogeneity, overall effect, overall *p*, and effect variance are derived from the random component of the meta-analysis. Minus sign of the overall effect indicates decreased value of the traits in the treatment (electrical stimulation) group, and vice-versa for plus sign.

Trait	Subgroups	No. of Studies	Raw Mean Difference	Effect Size	Weighted Mean Difference	95% CI	Overall *p*	*I*^2^ (%)
Ultimate pH	Low voltage	7	−0.07	−0.39	−0.04	(−0.69, 0.09)	<0.001	0
	Medium voltage	15	−0.12	−0.74	−0.03	(−1.29, −0.19)	<0.001	70
	High voltage	5	−0.09	−0.72	−0.04	(−1.12, 0.32)	<0.001	48
	**Overall effect**	**27**	**−0.09**	**−0.61**	**−0.03**	**(−0.84, −0.39)**	**<0.001**	**43**
WBSF (kg)	Low voltage	9	−1.02	−0.64	−0.98	(−1.15, −0.14)	<0.01	73
	Medium voltage	9	−1.06	−0.71	−1.05	(−1.12, −0.31)	<0.001	75
	High voltage	8	−1.09	−0.87	−1.05	(−1.38, −0.37)	<0.001	78
	Aged 1–3	16	−1.10	−0.78	−1.07	(−1.19, −0.38)	<0.001	85
	Aged 4–7	11	−1.06	−0.56	−1.04	(−1.02, −0.1)	<0.05	78
	Aged 8–14	10	−1.03	−0.61	−1.02	(−0.96, −0.26)	<0.001	71
	**Overall effect**	**63**	**−1.06**	**−0.7**	**−1.04**	**(−0.87, −0.53)**	**<0.001**	**78**
Sarcomere length (μm)	Low voltage	4	0.05	0.78	0.01	(−0.15, 1.70)	0.10	82
	Medium voltage	15	0.07	0.26	0.02	(−0.07, 0.59)	0.12	29
	High voltage	4	0.03	0.31	0.01	(−0.32, 0.95)	0.34	76
	**Overall effect**	**23**	**0.05**	**0.45**	**0.01**	**(0.10, 0.81)**	**<0.01**	**74**
CL (%)	Low voltage	7	−0.22	0.09	−0.02	(−0.18, 0.36)	0.52	0
	Medium voltage	15	−1.06	−0.4	−0.12	(−0.75, −0.06)	0.02	31
	High voltage	5	−0.67	0.01	−0.01	(−0.28, 0.29)	0.97	0
	Aged 1–3	21	−0.23	−0.05	−0.02	(−0.26, 0.16)	0.65	0
	Aged 4–7	8	−0.40	−0.14	−0.01	(−0.41, 0.13)	0.32	20
	Aged 8–14	5	−0.09	−0.27	−0.04	(−0.63, 0.10)	0.15	36
	**Overall effect**	**54**	**−0.45**	**−0.11**	**−0.03**	**(−0.22, −0.01)**	**<0.05**	**0**
Purge loss (%)	**Overall effect**	**4**	**−1.67**	**−0.61**	**−0.05**	**(−0.98, −0.25)**	**<0.001**	**14**
L*	Low voltage	6	2.01	0.64	0.18	(0.32, 0.97)	<0.001	8
	Medium voltage	15	1.14	0.51	0.4	(0.00, 1.03)	0.05	78
	High voltage	4	1.08	0.79	0.06	(0.28, 1.31)	<0.001	46
	Aged 1–3	11	1.01	0.55	0.01	(0.24, 0.85)	<0.001	61
	Aged 4–7	6	1.06	0.49	0.19	(0.00, 0.98)	<0.05	71
	**Overall effect**	**42**	**1.26**	**0.59**	**0.16**	**(0.41, 0.77)**	**<0.001**	**62**
a*	Low voltage	6	0.52	0.43	0.12	(0.16, 0.71)	<0.001	37
	Medium voltage	14	1.06	0.41	0.20	(0.12, 0.70)	<0.001	0
	High voltage	4	0.46	0.32	0.09	(−0.04, 0.68)	0.08	5
	Aged 1–3	10	0.62	0.41	0.07	(0.21, 0.62)	<0.001	4
	Aged 4–7	6	0.70	0.34	0.09	(0.09, 0.59)	<0.001	0
	**Overall effect**	**40**	**0.67**	**0.40**	**0.11**	**(0.28, 0.51)**	**<0.001**	**0**
b*	Low voltage	6	0.67	0.41	0.01	(0.06, 0.76)	<0.05	35
	Medium voltage	14	0.11	0.24	0.13	(−0.03, 0.51)	0.08	0
	High voltage	4	0.57	0.46	0.15	(0.10, 0.82)	<0.05	0
	Aged 1–3	10	0.38	0.37	0.05	(0.07, 0.67)	<0.05	49
	aged 4–7	6	0.31	0.22	0.06	(−0.04, 0.48	<0.05	0
	**Overall effect**	**26**	**0.37**	**0.34**	**0.08**	**(0.21, 0.47)**	**<0.001**	**20**
MFI	Low voltage	4	0.05	0.03	0.01	(−0.35, 0.42)	0.86	0
	Medium voltage	12	0.06	0.18	0.02	(−0.17, 0.52)	0.32	0
	High voltage	3	0.03	0.41	0.01	(−0.17, 0.98)	0.17	43
	Aged 1–3	7	0.03	0.05	0.02	(0.04, 0.86)	<0.05	61
	Aged 4–7	4	0.02	0.03	0.01	(−0.46, 0.52)	0.91	0
	**Overall effect**	**30**	**0.03**	**0.22**	**0.01**	**(0.06, 0.37)**	**<0.01**	**12**

Database: [2,4,5,8,9,30,32,33,34,35,36,37,38,39,40,41,42,43,44,45,46,47,48,49]. The overall effect of electrical stimulation for each parameter is given in bold.

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
