# Peer review of "A Meta-Analysis of the Effectiveness of High, Medium, and Low Voltage Electrical Stimulation on the Meat Quality of Small Ruminants"

_foods, 2020, doi:10.3390/foods9111587_

Round 1
Reviewer 1 Report
The conceptualisation of the MS was well performed and the choice of statistical tools was appropriate. the conclusions need to highlight more the output of the data gathered and how this study can be of use.
The MS was correctly designed, can you please explain further the choice of pH, Warner-Bratzler shear force, cooking loss, purge loss, sarcomere length , myofibrillar-fragmentation index as main factors to consider in this study
Did heterogeneity of the studies have a relevant impact on the results?
ES is a technology applied to increase meat tenderness and color., how you see the usefulness of the output of this study for a better understanding of the impact of EC on the meat quality. Please explain more in the conclusions
Reviewer 2 Report
General comments
This study is an interesting approach to the holistic interpretation of the scientific literature on the effects of electrical stimulation of small ruminant carcasses.
As the authors point out, this approach enables the discrepancies among individual studies to be minimised, while also offers a broad overview of the main trends described for each individual variable among a wide range of studies.
Although it is likely that the statistical studies carried out are beyond the command of mathematics of most readers, nevertheless, it represents a valuable attempt to standardize the extraordinarily varied conditions usually found among the numerous studies on the subject, in such a way that it allows a broad comparison of studies carried out under very diverse experimental conditions.
Under this perspective, the work is valuable and it is worth publishing.
However, there are some minor details that should be taken into account to improve the manuscript.
Specific comments
In addition to species, it would be necessary to talk about commercial categories or types, given that sheep and lamb can’t be classified as different species. Please, rewrite.
In this regard, it is also suggested that “goat kid”, replaces the category “goat”, taking into account the carcass weight mentioned in page 3, lines 111-112.
Page 1, line 37. The association between rigor mortis and the specific pH value mentioned is not as direct and unambiguous as the authors claim, given that it depends on different factors. Please, remove pH value.
Page 2, line 53. Reference to meta-analysis “in the field of science” is too generic, and even vague. Please, remove, or focus it on a more specific aspect or research area.
Page 2, lines 86-88. Although it is probable that the target audience is familiar with CIELAB meat colour parameters, a reference to the literature is missed (e.g., CIE, 1976)
Page 3, line 132. Please, define r.m.s. voltage at least as the effective value of AC.
Page 7, line 187. Please, define NES as non-electrically stimulated.
Page 10, line 238. Please, identify “thick” and “thin” filaments in sarcomeres as myosin and actin.
Page 10, line 245. The explanation for shear force is confusing. What is exactly meant by “… a balance between the sarcomere length ad the extent of postmortem proteolysis and collagen”? Given that collagen is also a protein, is proteolysis restricted to muscle protein fractions other than collagen? Please, rephrase.
Reviewer 3 Report
Manuscript foods-979565 is a review article that presents meta-analysis data of previous studies in the literature concerning the statistical proof of beneficial effects of electrical stimulation (ES) on meat quality of small ruminants in terms of ultimate pH, tenderness, enhanced proteolysis and higher colorimetric values. The paper is clearly and well written whereas the presented information is organized properly. In fact there is a novel aspect in this study involving the already existed literature in terms of statistical analysis of data and the revealing of new insights. I have some minor technical errors to comment on. These are:
Line 96 and elsewhere. ‘’described by [20]’’ is not properly given. Kindly provide authors name followed by et al.
-Table 1.Provide Table 1 in a better resolution.
